# Endothelial Cell Apoptosis but Not Necrosis Is Inhibited by Ischemic Preconditioning

**DOI:** 10.3390/ijms25021238

**Published:** 2024-01-19

**Authors:** Jarosław Zalewski, Marta Szajna, Konrad Stępień, Karol Nowak, Aleksandra Karcińska, Alicia del Carmen Yika, Kornelia Krawczyk, Krzysztof Karwat, Magdalena Zalewska, Piotr Pierzchalski

**Affiliations:** 1Department of Coronary Artery Disease and Heart Failure, Jagiellonian University Medical College, 31-202 Krakow, Poland; konste@interia.eu (K.S.); k.nowak.uj@gmail.com (K.N.); karwat.k@gmail.com (K.K.); 2John Paul II Hospital, 31-202 Krakow, Poland; martaszajna1@yahoo.com; 3Department of Thromboembolic Disorders, Jagiellonian University Medical College, 31-202 Krakow, Poland; 4Student Research Group, Department of Coronary Artery Disease and Heart Failure, Jagiellonian University Medical College, 31-202 Krakow, Poland; aleksandrakarcinska@gmail.com (A.K.); alicia.yika@student.uj.edu.pl (A.d.C.Y.); korneliakrawczyk3@wp.pl (K.K.); 5Department of Emergency Medicine, Faculty of Health Science, Jagiellonian University Medical College, 31-126 Krakow, Poland; magdalena.zalewska@uj.edu.pl; 6Faculty of Health Science, Jagiellonian University Medical College, 31-126 Krakow, Poland; piotr.pierzchalski@uj.edu.pl

**Keywords:** endothelium, ischemic preconditioning, hypoxia, reoxygenation, apoptosis

## Abstract

This study aimed to assess the influence of ischemic preconditioning (IP) on hypoxia/reoxygenation (HR)-induced endothelial cell (EC) death. Human umbilical vein endothelial cells (HUVECs) were subjected to 2 or 6 h hypoxia with subsequent reoxygenation. IP was induced by 20 min of hypoxia followed by 20 min of reoxygenation. Necrosis was assessed by the release of lactate dehydrogenase (LDH) and apoptosis by double staining with propidium iodide/annexin V (PI/AV), using TUNEL test, and Bcl-2 and Bax gene expression measured using RT-PCR. In PI/AV staining, after 24 h of reoxygenation, 30–33% of EC were necrotic and 16–21% were apoptotic. In comparison to HR cells, IP reduced membrane apoptosis after 24 h of reoxygenation by 50% but did not influence EC necrosis. Nuclear EC apoptosis affected about 15–17% of EC after 24 h of reoxygenation and was reduced with IP by 55–60%. IP was associated with a significantly higher Bcl-2/Bax ratio, at 8 h 2–4 times and at 24 h 2–3 times as compared to HR. Longer hypoxia was associated with lower values of Bcl-2/Bax ratio in EC subjected to HR or IP. IP delays, without reducing, the extent of HR-induced EC necrosis but significantly inhibits their multi-level evaluated apoptosis.

## 1. Introduction

The preconditioning process delays, without reducing, the extent of necrosis in endothelial cells exposed to hypoxia and reoxygenation. However, it significantly inhibits their multi-level evaluated apoptosis. The protective effect of the preconditioning process partially depends on the duration of the hypoxic phase.

The heart, along with every other organ, comprises a network of blood vessels. Thus, ischemia/reperfusion (I/R) injury affects elements of the vascular bed supplying the tissue subjected to I/R, including endothelial cells (ECs). I/R impairs endothelial function in large coronary arteries [1,2,3,4], arterioles with a diameter below 100 µm [5], microcirculatory vessels, and veins [6,7,8]. It also leads to structural damage of the coronary microcirculatory vessels, which, along with interstitial and cardiomyocyte edema [9], underlies the no-reflow phenomenon. Similarly, under in vitro conditions, ECs subjected to hypoxia/reoxygenation (HR) [10] or oxidative stress undergo damage.

In various experimental models, it has been demonstrated that ischemic preconditioning (IP) induces endogenous protective mechanisms against the consequences of prolonged I/R. Studies based on animal models suggest that IP can also protect the endothelium of epicardial coronary arteries in vivo [11,12,13,14] and microcirculatory vessels both in vivo [5,14] and in situ [15,16,17] against I/R-induced dysfunction. The IP process induces endothelial mechanisms that limit the inflammatory response triggered by I/R and prevent the expression of adhesion molecules on the endothelial surface, thereby inhibiting neutrophil adhesion [14,18,19,20]. Additionally, it reduces the release of endothelial cytokines, which in turn limits chemotaxis and the accumulation of inflammatory cells [16,18].

Preliminary clinical observations also seem to confirm that IP may prevent I/R-induced endothelial dysfunction and neutrophil activation in humans [21]. On the other hand, no effects of IP have been demonstrated in relation to the I/R of coronary circulation in dogs, where IP did not prevent either the reduction of resting perfusion within the myocardium or the decrease in coronary reserve between 30 min and 4 h of reperfusion [22]. 

Apoptosis is active, energy dependent, precisely regulated, influenced by various external factors, and a form of cell death [23,24]. Apoptosis is a multi-stage, time-distributed process involving characteristic changes in individual cell structures [25]. Within the circulatory system, this mechanism is observed in various scenarios, such as the apoptosis of cardiomyocytes during the maturation of the right ventricle in children, aging processes, or following myocardial infarction. It also occurs in conditions like hypertension, acute heart failure, post heart transplant, and in the smooth muscle or endothelium during ischemia/reperfusion (I/R) and atherosclerosis [26,27]. 

Classical IP, both in vivo and in vitro models, inhibits the process of cardiomyocyte apoptosis [28,29,30,31,32]. In rats, five repeated episodes of 5 min ischemia and reperfusion significantly reduce not only the infarct size but also the number of myocytes undergoing apoptosis [30]. In an in vivo rat heart I/R model, it has also been demonstrated that a single 5 min preconditioning stimulus preceding a 30 min ischemic period significantly reduces the percentage of apoptotic cells in the threatened infarct zone, from 29% in the control group to 3.5% in the preconditioned tissue. Meanwhile, epicardial artery ligation preceded by 5 min of ischemia and 5 min of reperfusion was associated with a significant reduction in cardiomyocyte apoptosis, as well as decreased neutrophil accumulation in the infarct area at risk in dogs [33]. In all the above-mentioned studies, IP resulted in a reduction of the infarct zone. However, it remains unknown as to what extent this reduction was attributed to the decrease in apoptotic cell numbers or the prevention of the transition of apoptotic cells into necrosis.

In contrast to cardiomyocytes, we do not know whether and how quickly ECs undergo HR-induced necrosis and apoptosis. Moreover, it has not yet been clearly established whether IP can protect the viability of EC subjected to HR or I/R [34,35]. IP modifies the expression of proteins involved in the process of apoptosis. The increase in the Bcl-2/Bax ratio is associated with the inhibition of cytochrome C release from mitochondria stimulated with an apoptosis-inducing stimulus [36,37]. In the I/R model of the isolated rat heart, IP inhibits the apoptosis process by increasing the expression of Bcl-2 and inhibiting the expression of p53 [31] and Bax [29]. In turn, Nakamura et al. showed that in the rat heart subjected to I/R, IP prevents myocyte apoptosis, which is accompanied by a reduction in Bax expression without changes in Bcl-2 expression [29]. In Zhao’s study [38], the increasing number of apoptotic cells during reperfusion was accompanied by an increase in the expression of Bax and p53 and a decrease in the expression of Bcl-2.

The aim of this work is to describe the process of endothelial cell death induced by HR and to determine the impact of the IP process on the course of HR-induced EC death. The in vitro HR model is intended, with some simplification, to imitate the I/R processes occurring in vivo, providing the possibility of precisely verifying the working hypothesis.

## 2. Results

### 2.1. Lactate Dehydrogenase Release

There were no differences between the studied groups at baseline. EC did not release LDH during hypoxia (Table 1). The LDHi in the control group, both after a 2 and 6 h period of hypoxia, increased gradually from 0.017 at baseline to 0.092–0.110 after 24 h of reoxygenation. The most dynamic LDH release was in the first 2 h of reoxygenation both in HR and in IP groups. After 2, 8, and 24 h of reoxygenation, there was a significant difference in the LDHi between the control group and HR or IP cells, both after 2 and 6 h hypoxia. After 8 h of reoxygenation, the LDHi was significantly lower in the IP versus HR group, regardless of the hypoxia time. The latter effect was transient, and after 24 h of reoxygenation, both in groups of HR and IP, the LDHi—regardless of the duration of the hypoxia phase—was more than three times higher than in the control group (Table 1). 

The length of the hypoxia influenced the amount of LDH release both in HR and IP groups. In HR group, the longer 6 h hypoxia significantly increased LDH release after the first 2 (*p* < 0.01) and after 24 h of reoxygenation, while in IP cells it increased after 2 (*p* < 0.001) and 8 (*p* < 0.01) h of reoxygenation (Table 1).

### 2.2. Endothelial Cell Viability in Double Staining with Propidium Iodide and Annexin V

At baseline, after 2 or 6 h hypoxia, as well as after 2 h reoxygenation, irrespective of the hypoxia time, the percentage of viable (V), apoptotic (A), and necrotic (N) cells was identical in the three studied groups.

After 8 h of reoxygenation, both in the HR and IP groups, regardless of the duration of the hypoxia phase, the percentage of necrotic or apoptotic cells was significantly higher compared to control conditions (Table 2). Simultaneously, the percentage of necrotic or apoptotic cells in the HR groups was higher than in the IP cells. Differences in the percentage of viable, apoptotic, and necrotic cells between the HR or IP versus the control group persisted up to 24 h of reoxygenation, regardless of the duration of the hypoxic phase. After 24 h of reoxygenation, in the IP groups—regardless of the duration of the hypoxia phase—the percentage of apoptotic cells was significantly lower compared to HR cells. However, IP did not reduce the percentage of necrotic cells after 24 h of reoxygenation as compared to the HR group. The dynamics of an increase in apoptotic and necrotic cells, both in the HR as well as IP groups, were the highest between the 2nd and 8th hour of reoxygenation. 

The length of the hypoxia phase influenced the percentage of apoptotic cells in both the IP and HR groups. In IP and HR conditions, both after 8 and 24 h of reoxygenation, longer hypoxia was associated with a significantly higher percentage of apoptotic cells (*p* < 0.05 for all comparisons). The length of the hypoxia phase did not affect the percentage of necrotic cells during the entire reoxygenation phase. Appendix A shows an example of double staining with PI/AV.

### 2.3. Endothelial Cell Apoptosis in the TUNEL Technique 

At baseline, after a 2 or 6 h hypoxia, as well as after a 2 or 8 h reoxygenation phase—regardless of the duration of the hypoxia phase—there were no differences in the percentage of cells with signs of apoptosis in the cell nucleus identified based on the TUNEL technique in all study groups (Figure 1).

After 24 h of reoxygenation—regardless of the hypoxia time—the percentage of TUNEL-positive cells in the IP group was almost three times higher, and in the HR group, it was over six times higher than in the control group. Moreover, in the HR groups, there were twice more TUNEL-positive HUVECs than in the IP group. The hypoxia time had no impact on the percentage of TUNEL-positive apoptotic cells. Appendix A shows an example of TUNEL staining.

### 2.4. Expression of β-Actin, Bax, Bcl-2, e-NOS, i-NOS, and Cox-2

Gene expression of β-actin (ACTB), Bax, Bcl-2, e-NOS, i-NOS, Cox-2, and the ratio of Bcl-2/Bax was assessed in endothelial cells at baseline (Figure 2A), after 8 h (Figure 2, Figure 3 and Figure 4), and after 24 h (Figure 2, Figure 3 and Figure 4) of reoxygenation. The ACTB expression remained stable throughout the experiment (Figure 2B).

After 8 h of reoxygenation, there was a significant increase in Bax expression in both IP groups (IP2, IP6) and HR groups (HR2, HR6) compared to the control ECs (Figure 3A). No differences in Bax expression were observed between groups with different hypoxia times (HR2 vs. HR6, IP2 vs. IP6). IP did not influence Bax expression in groups with the same length of hypoxia (HR2 vs. IP2, HR6 vs. IP6). At 24 h of reoxygenation, Bax expression in groups HR2, HR6, IP2, and IP6 was still higher than in the control group, but IP was associated with a significant reduction of Bax compared to the respective HR groups (IP2 vs. HR2, IP6 vs. HR6).

The Bcl-2 expression at the 8th hour of reoxygenation in the HR2 and HR6 groups was comparable with control ECs, while in both IP groups, it was significantly higher than in the control group and in groups with the same hypoxia time (HR2 and IP2, HR6 vs. IP6) (Figure 3B). Moreover, in IP groups, Bcl-2 expression was higher when the hypoxia time was shorter (IP2 vs. IP6). After 24 h of reoxygenation, Bcl-2 expression was significantly higher in both groups subjected to HR or IP compared to the control conditions. Bcl-2 expression was significantly higher when groups with the same hypoxia time were preconditioned (HR2 vs. IP2, HR6 vs. IP6) and when the IP process was preceded by shorter hypoxia (IP2 vs. IP6).

The Bcl-2/Bax ratio at the eighth hour of reoxygenation was lower in both HR groups (HR2, HR6) and higher in both IP groups (IP2, IP6) compared to the control ECs (Figure 2C). No significant differences were observed in the Bcl-2/Bax ratio between groups subjected to hypoxia at different times (HR2 vs. HR6). However, a shorter hypoxia time was associated with a higher value of the Bcl-2/Bax ratio in the IP groups. Moreover, the IP process was associated with a higher value of the Bcl-2/Bax ratio in groups with the same hypoxia time (HR2 vs. IP2, HR6 vs. IP6). At 24 h of reoxygenation, a shorter hypoxia time was associated with a higher value of the Bcl-2/Bax ratio in groups subjected to hypoxia (HR2 vs. HR6) or IP (IP2 vs. IP6). The value of the Bcl-2/Bax ratio in the IP groups was higher compared to the HR groups (IP2 vs. HR2, IP6 vs. HR6).

The expression of e-NOS at the eighth hour of reoxygenation was lower following 6 h hypoxia, and this change was prevented by the IP process (Figure 4A). No differences in e-NOS expression were observed between groups with different hypoxia times (HR2 vs. HR6, IP2 vs. IP6). At 24 h of reoxygenation, the expression of e-NOS was significantly lower in both groups subjected to hypoxia compared to control ECs (HR2, HR6), while again these changes were prevented by the IP process. Similarly, at the 8 h reoxygenation mark, no differences in e-NOS expression were observed between groups with different hypoxia times (HR2 vs. HR6, IP2 vs. IP6). I-NOS expression was not detected in HR2, HR6, IP2, and IP6 conditions either after 8 or 24 h of reoxygenation.

At the eighth hour of reoxygenation, the expression of Cox-2 in both HR groups was comparable with control ECs. Simultaneously, in both IP groups, Cox-2 expression was higher compared to the control conditions (Figure 4B), as well as when compared with HR groups with the same hypoxia time (HR2 vs. IP2, HR6 vs. IP6).

### 2.5. HUVEC Morphological Changes Following Hypoxia/Reoxygenation

Phase-contrast microscopy did not reveal changes in HUVEC morphology during hypoxia (Appendix A). However, following reoxygenation, the endothelial cells (ECs) underwent shrinkage, and their shape transformed from polygonal to irregular. As reoxygenation progressed, increasing intercellular spaces became apparent due to cell shrinkage and detachment from the dish bottom. 

Under the influence of hypoxia, actin filaments underwent reorganization, and stress fibers were either lost or shortened (Appendix A). After 8 h of reoxygenation, F-actin accumulated in the perinuclear space, while stress fibers were degraded. Subsequently, after 24 h of reoxygenation, the effects of hypoxia and the early reoxygenation phase were reversed, and stress fibers were rebuilt.

## 3. Discussion

This study reveals that the Ischemic Preconditioning (IP) process, while delaying endothelial cell necrosis, does not mitigate its extent; however, it effectively inhibits the intricate process of apoptosis in Human Umbilical Vein Endothelial Cells (HUVEC) subjected to hypoxia and reoxygenation. The injury induced by Ischemia/Reperfusion (I/R) in endothelial cells within the myocardium underlies the no-reflow phenomenon, the reperfusion inflammatory response, and the expansion of the infarct zone. Consequently, enhancing endothelial cell viability through IP may potentially contribute to improved endothelial function, safeguarding coronary perfusion during I/R, and augmenting the protective effects of cardiomyocyte preconditioning.

Our findings reveal that the IP process substantially reduces, by approximately 3–4 times, the loss of cell membrane integrity in HUVEC between 2 and 8 h of reoxygenation. However, this protective effect is transient, as the percentage of viable ECs, both conditioned and unconditioned, becomes comparable after 24 h of reoxygenation. Hence, the conditioning process temporarily retards but does not arrest the decline in cell membrane integrity in ECs.

A similar trend is observed concerning the rate of necrotic cell formation in double-staining with Propidium Iodide (PI) and Annexin V (AV). IP reduces HR-induced cell necrosis 2–3 times during the first 8 h of reoxygenation. Yet, in the subsequent hours of reoxygenation, the rate of necrotic cell formation in HUVEC subjected to IP accelerates. Ultimately, after 24 h of reoxygenation, no differences are observed, indicating that IP only delays the irreversible impairment of the cell membrane in HUVEC.

In contrast to necrosis, the apoptosis process in HUVEC is modulated by preconditioning. The IP process does not completely halt but significantly reduces the percentage of cells displaying features of membrane apoptosis after 8 h, persisting after 24 h of reoxygenation. The rate of apoptotic cell increase in the conditioned population is fastest between 2 and 8 h of reoxygenation, akin to non-conditioned cells. However, compared to non-conditioned cells, this rate is approximately twice slower throughout the entire reoxygenation phase. Furthermore, in cells subjected to IP, apoptosis at the nuclear level begins between 8 and 24 h of reoxygenation; however, IP distinctly limits this process by more than half.

The lack of standardized experimental conditions poses a significant challenge in preconditioning studies using in vitro models [39]. Nevertheless, in vitro and in vivo studies on IP mediators and mechanisms consistently yield comparable results. The outcomes of conditioning for a specific organ result from the interplay of IP mechanisms in various cell types, as evidenced by in vitro studies.

The benefits of a cell culture model in preconditioning studies include the ability to study human cells, perform genetic modifications, manipulate during hypoxia and reoxygenation, measure channel activity and redox potential, separately assess necrosis and apoptosis, investigate the impact of external environment changes, and control cellular interactions. However, drawbacks include uncertainties about experimental condition stability and a lack of data on how well the model mimics in situ or in vivo I/R or infarction.

In this study, the IP stimulus was a single episode of 20 min hypoxia followed by 20 min of reoxygenation. Short episodes of ischemia (up to 15 min) have been extensively studied in coronary artery endothelial dysfunction [40,41]. Also, different experimental IP protocols have been tested in cell culture so far. Zhou et al. showed that 10 min of anoxia and 10 min of reoxygenation protected coronary endothelial cells against anoxia and reoxygenation injury [42]. Moreover, IP with 1 h anoxia and 1 h reoxygenation of cultured rat aortic endothelial cells prevented reoxygenation-induced, free-radical-mediated expression of ICAM-1 involving activation of protein kinase C and the production of nitric oxide and free radicals, and this was associated with a lesser adhesion of neutrophils to endothelial cells [43]. In turn, in vivo IP with 10 min occlusion and reperfusion preceding 60 min ischemia and 120 min of reperfusion reduced endothelial dysfunction of coronary arterioles [5]. Also, the results of our unpublished preliminary experiments showed that IP induced by 20, 30, or 60 min of hypoxia with, respectively, 20, 30, or 60 min of reoxygenation reduced LDH release following a prolonged 6 h hypoxia and 6 h of reoxygenation. The results presented here suggest that an IP protocol with 20 min of hypoxia followed by 20 min of reoxygenation was effective in inducing conditioning mechanisms.

The IP process did not prevent the increase in Bax expression during reoxygenation and did not alter Bax expression after 8 h compared to non-conditioned cells. However, after 24 h, conditioning led to a significant reduction in Bax expression compared to HUVEC not subjected to IP. Throughout the reoxygenation period, conditioned cells exhibited a significant increase in Bcl-2 expression compared to the control and hypoxia-exposed cells, resulting in a considerable enhancement of intracellular protective potential, as indicated by the Bcl-2/Bax ratio.

The key role in the development of reperfusion-induced apoptosis is attributed to the interaction between blood cells and endothelium [44] mediated by the cytokines and reactive oxygen species released from activated neutrophils, endothelial cells, and myocytes. Endothelial cells in the studied in vitro model were not subject to such an interaction with blood cells, including neutrophils, which may be associated with the lack of induction of i-NOS expression. However, based only on the lack of i-NOS expression, it is difficult to answer whether the inflammatory process in cultured HUVEC was not activated when the more that the expression of Cox-2 increased following 2 h hypoxia and 24 h reoxygenation and even more as a consequence of the IP process. Basically, cox-2 expression accompanies harmful processes such as hypersensitivity, cancer, sepsis, apoptosis, and also inflammatory processes. On the other hand, the constant prostaglandins production by Cox-2 was found to be associated with late IP cardioprotection [45] and may also inhibit the apoptosis process in various cells [46].

The tissue hypoxia inhibits the expression of more than 95% of genes as compared to the baseline, and only a few genes increase their expression. McQuillan et al. [47] showed that in endothelial cells that underwent hypoxia, e-NOS expression was reduced by 40–60%, and this effect lasted for at least 48 h. In turn, 24 h anoxia followed by 3 h reoxygenation of human endothelial coronary cells was associated with a significant reduction of the expression, protein concentration, and activity of e-NOS [48]. In our experiment, after 24 h of reoxygenation, there was no change in e-NOS expression in the control group indicating a stable genotype of HUVECs despite the lack of shear stress, while the reduction of e-NOS expression following hypoxia/reoxygenation was prevented by the IP process.

A key limitation of our cell culture model is the absence of shear stress, a crucial factor for endothelial cell function in vivo [49,50]. The static cell culture conditions do not subject endothelial cells to flow, impacting the model’s ability to respond to shear stress effects. Additionally, the in vitro environment lacks neural stimulation and the in vivo hormonal environment necessary for regulating endothelial function. The study was unable to investigate vasodilatory endothelial function under in vitro conditions.

## 4. Materials and Methods

### 4.1. Isolation, Identification, and Culture of Endothelial Cells 

Endothelial cells were isolated from the umbilical vein using the enzymatic method after natural delivery. The harvested cells were cultivated at 37 °C in a 5% carbon dioxide and 95% air environment in human endothelial serum-free medium (Gibco by Thermo Fisher, Waltham, MA, USA). The medium contained 5% fetal bovine serum (FBS, Gibco by Thermo Fisher, Waltham, MA, USA) and was supplemented with human recombinant epithelial growth factor (Gibco by Thermo Fisher, Waltham, MA, USA). The purity of endothelial cells (ECs) was verified through immunofluorescent staining against von Willebrand factor. An illustrative example of this staining is provided in Appendix A.

HUVECs in culture underwent routine assessments every 24 h. Upon reaching confluence, cells were passaged. To mitigate the potential impact of variations between cells from different umbilical cords on experimental outcomes, cells from diverse culture vessels were combined before centrifugation. For experiments, fully confluent ECs from the third passage were utilized.

### 4.2. Hypoxia and Reoxygenation in HUVEC Culture

Hypoxia induction in HUVECs involved filling the culture dish with deoxygenated Tyrod’s fluid. During the subsequent reoxygenation phase, the cells were placed in normally oxygenated Tyrod’s fluid, maintaining an oxygen content of 72 ± 3%. The control group cells were incubated in Tyrod’s fluid with an identical oxygen content. Throughout the experiment, culture dishes were housed in an incubator set at a constant temperature of 37 °C, with a humidity level of 98%, and precise concentrations of oxygen, nitrogen, and carbon dioxide (IG 750 Incubator, JOUAN, Saint-Herblain, France). During hypoxia, the oxygen concentration around the culture dish was sustained below 0.5%, and during reoxygenation, it was restored to atmospheric levels.

Tyrod’s fluid, serving as the medium to sustain hypoxic conditions, was saturated with 100% nitrogen two hours before each experiment. The medium was buffered to a pH of 7.4 ± 0.05, and its osmolarity was adjusted to 290 ± 5 mOsm. Oxygen content in Tyrod’s fluid, after 15 min of nitrogen saturation, was 10 ± 2%, gradually decreasing to 7 ± 2% after 30 min, 5 ± 1% after 1 h, and reaching 3 ± 1% after 2 h. Changes in oxygen content during nitrogen saturation and the hypoxic phase are depicted in Figure 5.

The oxygen saturation of Tyrod’s fluid within the tightly closed culture dish fluctuated over time. Based on oxygen content measurements using an oxygen sensor (Oxygen Meter CO-315, Elmetron, Zabrze, Poland), it was determined that oxygen saturation during the hypoxia phase increased from 3 ± 1% initially to 5 ± 1% after 20 min, 8 ± 2% after 1 h, 11 ± 2% after 2 h, 13 ± 3% after 4 h, and 15 ± 3% after 6 h. Osmolarity of the solutions used for culture and experiments was measured using an osmometer (Osmometer, Elmetron, Zabrze, Poland), and pH was determined with a pH-meter (pH-meter P-500, Consort, Monheim, Germany).

### 4.3. Study Protocol

Cells from the third passage were randomly allocated to one of five study groups (Figure 6): 

1. [C] Control Group: Cells were incubated with normally oxygenated Tyrod’s fluid.

2. [HR2] 2 h Hypoxia Followed by Reoxygenation: Cells were exposed to 2 h of hypoxia by incubation with deoxygenated Tyrod’s fluid, followed by reoxygenation in normally oxygenated Tyrod’s fluid. 

3. [HR6] 6 h Hypoxia Followed by Reoxygenation: Cells underwent 6 h of hypoxia with deoxygenated Tyrod’s fluid, followed by reoxygenation in normally oxygenated Tyrod’s fluid.

4. [IP2] IP Followed by 2 h Hypoxia and Reoxygenation: Cells experienced 20 min of hypoxia and 20 min of reoxygenation (IP), followed by 2 h of hypoxia and reoxygenation. After IP induction, cells were incubated for 2 h with deoxygenated Tyrod’s fluid and then in normally oxygenated Tyrod’s fluid.

5. [IP6] IP Followed by 6 h Hypoxia and Reoxygenation: Cells underwent 20 min of hypoxia and 20 min of reoxygenation (IP), followed by 6 h of hypoxia and reoxygenation. After IP induction, cells were incubated for 6 h with deoxygenated Tyrod’s fluid and then in normally oxygenated Tyrod’s fluid.

### 4.4. Endothelial Cell Death Measurement by Lactate Dehydrogenase Release

Lactate dehydrogenase (LDH) activity was assessed using the colorimetric LDH Cytotoxicity Detection kit (Takara Biomedicals, MK401, Takara Bio Inc., Shiga, Japan), employing a two-step enzymatic reaction. The LDH index (LDHi), representing the extent of cell damage, was calculated as LDHi = (AN − AC)/(AT − AC), where AN is the absorbance of the tested sample supernatant, AT is the absorbance of the sample incubated with Triton X-100 (Sigma-Aldrich, Burlington, MA, USA) which induces release of the entire cellular LDH pool, and AC is the absorbance of the reaction mixture. This kit enables the detection of 20–200 dead cells in a 200 µL volume.

### 4.5. Endothelial Cell Necrosis/Apoptosis in Double Staining with Propidium Iodide and Annexin V 

Staining involved the use of 1 µg/mL propidium iodide and 0.2 mg/mL Annexin V from the Annexin V Kit (Santa Cruz Biotechnology Inc., Dallas, TX, USA). Following the experiment, cells were washed, trypsinized, and stained. Flow cytometry analysis was conducted using a FACS Calibur flow cytometer (Beckton Dickinson, Franklin Lakes, NJ, USA) with appropriate negative controls.

### 4.6. Endothelial Cell Apoptosis in the TUNEL Technique

The MEBSTAIN Apoptosis Kit Direct assay (Medical & Biological Laboratories Co., Tokyo, Japan) was employed following the manufacturer’s instructions. Fixed cells were subjected to TdT buffer II/FITC-dUTP/TdT reaction, and fluorescence microscopy was used for observation. Negative controls included a sample with a mixture of TdT buffer II and FITC-dUTP.

### 4.7. Reverse Transcription Polymerase Chain Reaction (RT-PCR)

Total cellular RNA was isolated using TRIzol reagent (Gibco by Thermo Fisher, Waltham, MA, USA), and the concentration was determined using spectrophotometry. Both reverse transcription (RT) and PCR were performed using the Enhanced Avian HS RT-PCR Kit (Sigma-Aldrich, Burlington, MA, USA, MB-955). Specific primers (listed below) were used, and electrophoresis of the cDNA amplification product was conducted on an agarose gel. Gel images were scanned, and band intensity was quantified using a laser densitometer.

Primers:-ACTB: s-5′-AGCGGGAAATCGTGCGTG-3′, a-5′-GGGTACATGGTGGTGCCTG-3′-Cox-2: s-5′-CCGGACAGGATTCTATGGAGA-3′, a-5′-CAATCATCAGGCGACAGGAGG-3′-Bax, Bcl-2, i-NOS, e-NOS: Purchased from R&D Systems Inc. (Minneapolis, MN, USA).

Appendix A include phase-contrast microscope images, photos from a fluorescence microscope of von Willebrand factor, propidium iodide/annexin V preparations, DNA sticky ends in TUNEL technique, and actin cytoskeleton.

### 4.8. Statistical Analysis

Statistical analyses were conducted using SPSS Statistics software (Version 29.0.0.0, IBM Corp., Armonk, NY, USA). Normally distributed continuous variables were presented as mean ± standard deviation. Continuous variables were compared using Student’s *t*-test or ANOVA with post hoc Bonferroni correction. A two-tailed *p*-value of <0.05 was considered statistically significant.

## 5. Conclusions

Apoptosis and necrosis induced by hypoxia and reoxygenation in endothelial cells exhibit dissociation in terms of both time and cell ultrastructure. The preconditioning process delays, without reducing, the extent of necrosis in endothelial cells exposed to hypoxia and reoxygenation. However, it significantly inhibits their multi-level evaluated apoptosis. The protective effect of the preconditioning process partially depends on the duration of the hypoxic phase.

## Figures and Tables

**Figure 1 ijms-25-01238-f001:**
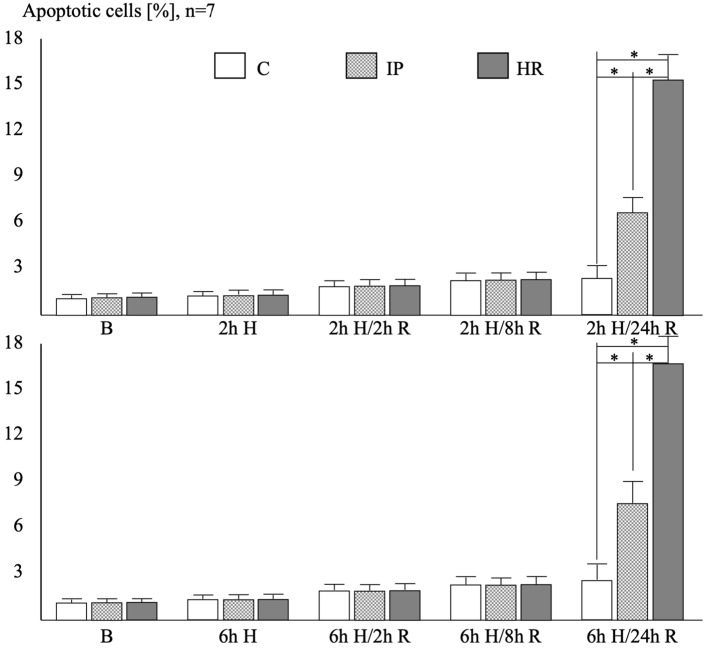
The percentage of apoptotic cells in the TUNEL test. Abbreviations: C: control group, IP: preconditioning, HR: hypoxia/reoxygenation, B: baseline conditions, H: hypoxia, R: reoxygenation, n: number of experiments. Post hoc differences: * *p* < 0.001.

**Figure 2 ijms-25-01238-f002:**
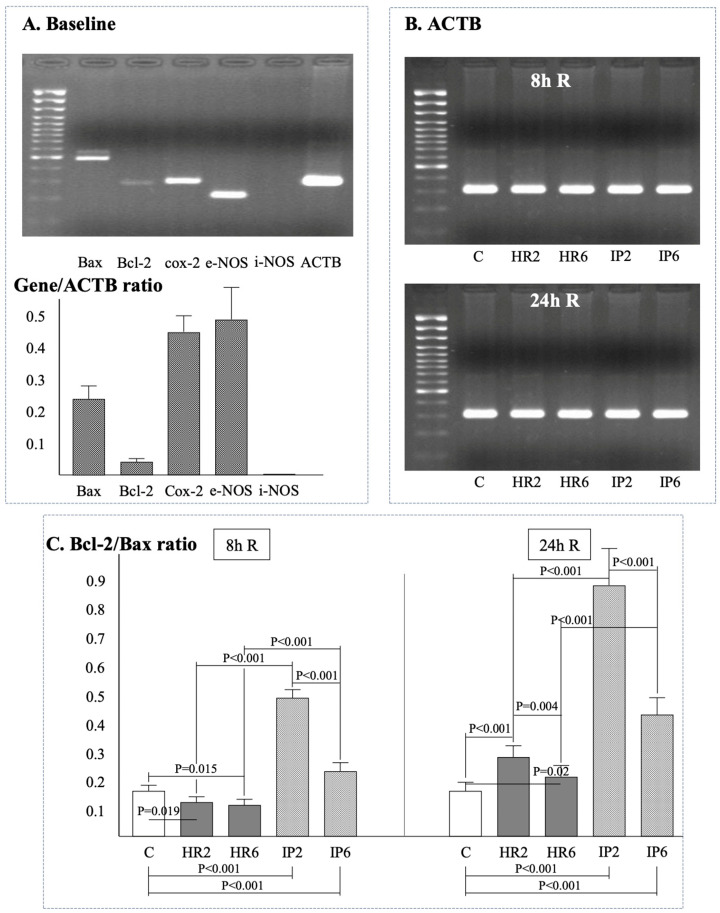
The expression of selected genes. (**A**) Baseline expression, (**B**) ACTB expression after 8 and 24 h of reoxygenation and (**C**) Bcl-2/Bax ratio after 8 and 24 h of reoxygenation. Abbreviations: R: reoxygenation, C: control group, HR2: 2 h hypoxia, HR6: 6 h hypoxia, IP2: 2 h hypoxia preceded by preconditioning, IP6: 6 h hypoxia preceded by preconditioning.

**Figure 3 ijms-25-01238-f003:**
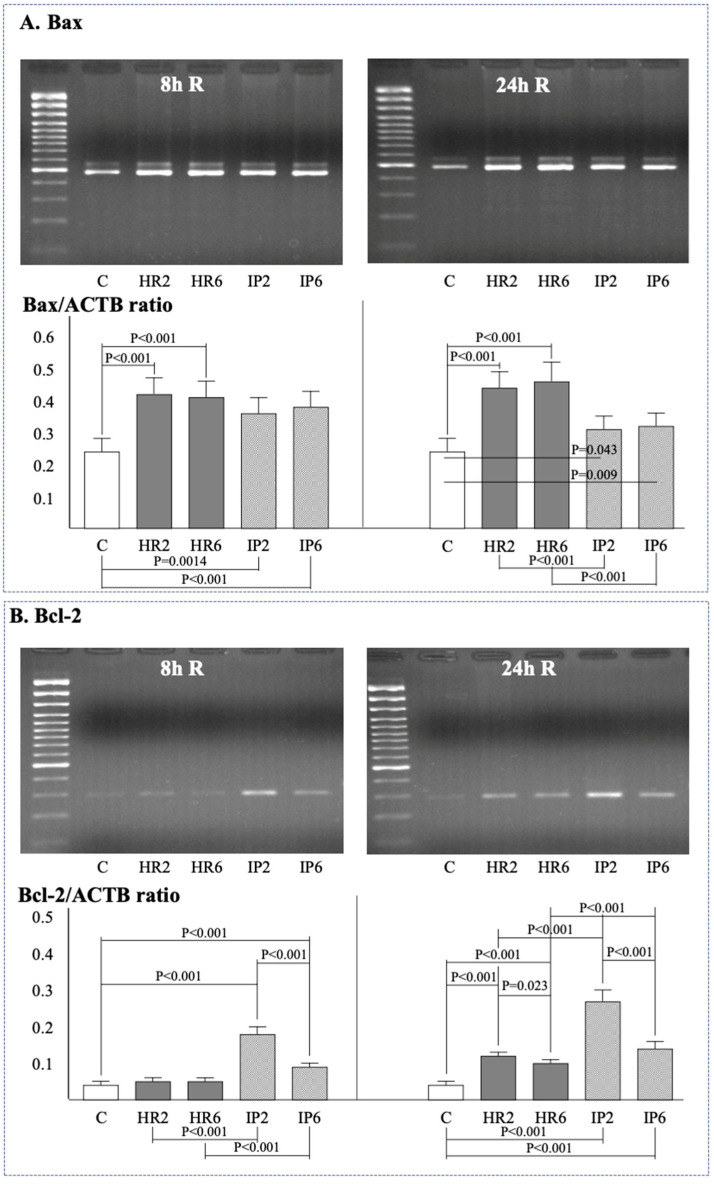
The expression of Bax after 8 and 24 h of reoxygenation (**A**) and Bcl-2 after 8 and 24 h of reoxygenation (**B**). Abbreviations: see Figure 2.

**Figure 4 ijms-25-01238-f004:**
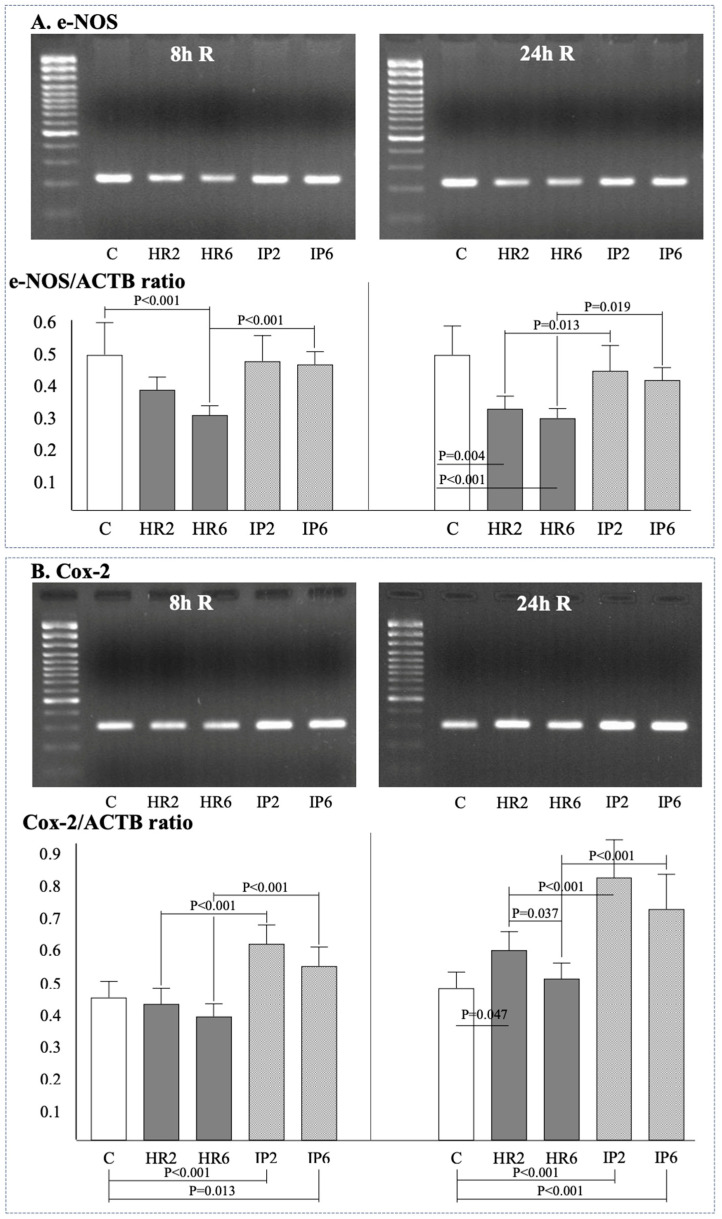
The expression of e-NOS after 8 and 24 h of reoxygenation (**A**) and of cox-2 after 8 and 24 h of reoxygenation (**B**). Abbreviations: see Figure 2.

**Figure 5 ijms-25-01238-f005:**
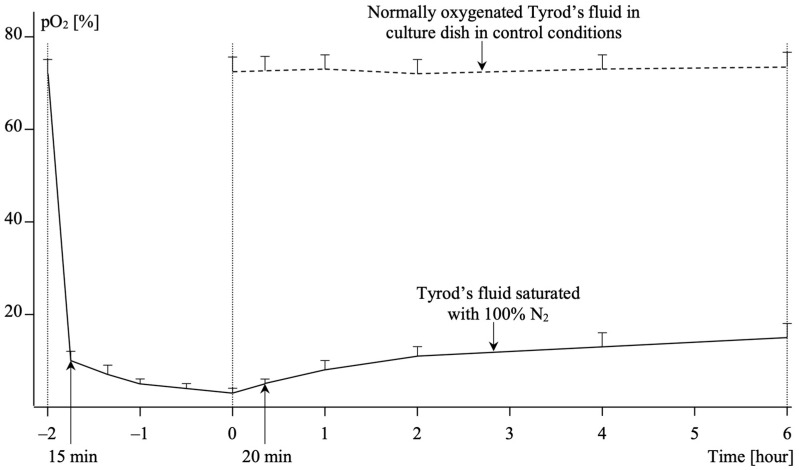
The oxygen saturation of Tyrod’s fluid.

**Figure 6 ijms-25-01238-f006:**
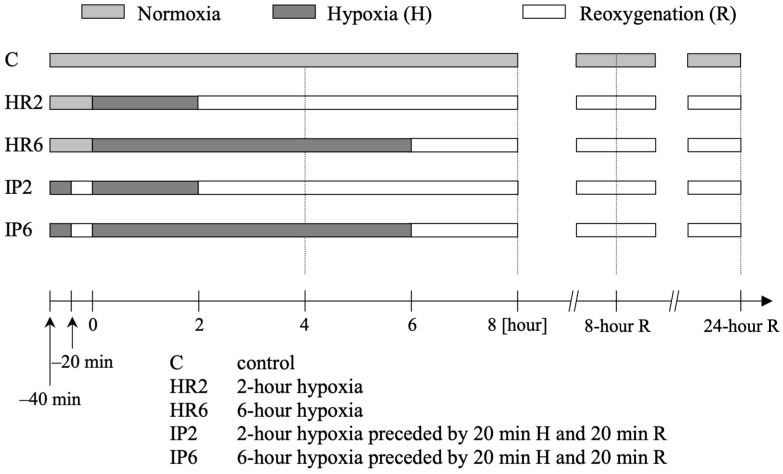
Experimental protocol.

**Table 1 ijms-25-01238-t001:** The lactate dehydrogenase index.

*n* = 12	Study Group	Baseline	H	2 h R	8 h R	24 h R
2 h hypoxia	C	0.017 ± 0.010	0.019 ± 0.011	0.023 ± 0.028	0.043 ± 0.019	0.092 ± 0.016
	IP	0.018 ± 0.011	0.020 ± 0.011	0.075 ± 0.021 *	0.096 ± 0.028 *^,^**	0.280 ± 0.052 *
	HR	0.016 ± 0.009	0.019 ± 0.012	0.098 ± 0.027 *	0.167 ± 0.033 *	0.305 ± 0.048 *
6 h hypoxia	C	0.017 ± 0.011	0.018 ± 0.011	0.035 ± 0.021	0.054 ± 0.027	0.110 ± 0.030
	IP	0.017 ± 0.010	0.018 ± 0.012	0.117 ± 0.023 *	0.134 ± 0.034 *^,^**	0.330 ± 0.079 *
	HR	0.018 ± 0.011	0.020 ± 0.011	0.146 ± 0.045 *	0.193 ± 0.043 *	0.370 ± 0.084 *

Abbreviations: C: control group, IP: preconditioning, H: hypoxia, R: reoxygenation, *n*: number of experiments. Post hoc differences: * *p* < 0.001 vs. control group, ** *p* < 0.001 vs. HR group.

**Table 2 ijms-25-01238-t002:** The percentage of viable, apoptotic and necrotic cells in double staining with propidium iodide and annexin V.

*n* = 6		2 h Hypoxia			6 h Hypoxia		
		C	IP	HR	C	IP	HR
8 h R	Apoptotic	2.3 ± 0.3	4.5 ± 0.5 *^,^**	6.5 ± 0.85 *	2.4 ± 0.3	5.4 ± 0.65 *^,^**	7.9 ± 0.9 *
	Necrotic	5.8 ± 0.7	9.1 ± 1.0 **^,^***	17.5 ± 2.4 *	6.2 ± 0.8	11.1 ± 1.6 **^,^***	20.2 ± 3.0 *
	Viable	91.9 ± 1.5	86.4 ± 2.8	76.0 ± 4.8	91.4 ± 1.6	83.5 ± 2.7	72.1 ± 4.9
24 h R	Apoptotic	2.4 ± 0.35	7.4 ± 0.9 *^,^**	15.6 ± 2.0 *	2.6 ± 0.4	10.6 ± 1.3	20.7 ± 2.2
	Necrotic	7.6 ± 0.91	26.9 ± 3.1 *	30.2 ± 4.0 *	8.8 ± 1.1	28.2 ± 3.3 *	33.2 ± 4.3 *
	Viable	90.0 ± 1.8	65.7 ± 3.8	54.2 ± 6.2	88.6 ± 1.8	61.2 ± 4.6	46.0 ± 6.5

Abbreviations: C: control group, IP: preconditioning, HR: hypoxia/reoxygenation, R: reoxygenation, *n*: number of experiments. Post hoc differences: * *p* < 0.001 vs. control group, ** *p* < 0.001 vs. HR group, *** *p* < 0.01 vs. control group.

## Data Availability

Data is contained within the article and Appendix A.

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
