# Peer review of "Endothelial Cell Apoptosis but Not Necrosis Is Inhibited by Ischemic Preconditioning"

_ijms, 2024, doi:10.3390/ijms25021238_

Round 1
Reviewer 1 Report
Comments and Suggestions for Authors
The manuscript is clear, concise, and easily understandable. However, during the review, I identified some inconsistencies, notably throughout the introduction, the Bax protein is written both in upper and lower case, please confirm if it is on purpose. In section 2.1 second paragraph, in the second phrase, when you mention “In IP cells, longer 6-hour hypoxia significantly increased LDH release after first 2 and after 24 hours of reoxygenation” I believe it is meant to be the HR cells. Also, in the same section, in the table, there is a study group named IR that I believe is supposed to be the HR. Figure 1 is missing the legend. The quality of the graphs presented in Figure 2 could also be significantly improved, since now it is presented in a quite unorganized manner, e.g. the title of the y-axis is presented on top in very small font and the p-values are scattered. Please revise the gene nomenclature throughout the whole manuscript as very often protein rather than gene names are presented, e.g. beta-actin is not a gene name, it should read ACTB.
Please address these concerns to enhance the overall quality of your manuscript.
Author Response
Thank you for the positive feedback and all comments.
- The Bax protein is written both in upper and lower case, please confirm if it is on purpose.
The spelling of studied genes has been corrected.
- In section 2.1 second paragraph, in the second phrase, when you mention “In IP cells, longer 6-hour hypoxia significantly increased LDH release after first 2 and after 24 hours of reoxygenation” I believe it is meant to be the HR cells. Also, in the same section, in the table, there is a study group named IR that I believe is supposed to be the HR.
Indicated errors have been improved.
- Figure 1 is missing the legend.
Figure 1 legend has been updated.
- The quality of the graphs presented in Figure 2 could also be significantly improved, since now it is presented in a quite unorganized manner, e.g. the title of the y-axis is presented on top in very small font and the p-values are scattered.
The whole Figure 2 has been reorganized and divided into three new figures.
- Please revise the gene nomenclature throughout the whole manuscript as very often protein rather than gene names are presented, e.g. beta-actin is not a gene name, it should read ACTB.
As suggested, gene names have been checked and rephrased.
Reviewer 2 Report
Comments and Suggestions for Authors
Review Reports
A brief summary of Manuscript ID: ijms-2802337
Endothelial cell apoptosis but not necrosis is inhibited by is-2 chemic preconditioning
This manuscript aimed to assess the influence of ischemic preconditioning (IP) on hypoxia/reoxygenation (HR)-induced endothelial cell (EC) death. The in vitro HR model is intended, with some simplification, to imitate the I/R processes occurring in vivo, providing the possibility of precisely verifying the working hypothesis. Human umbilical vein endothelial cells (HUVEC) were subjected to 2- or 6-hour hypoxia with subsequent reoxygenation. IP was induced by 20 minutes of hypoxia followed by 20 minutes of reoxygenation. This study reveals that the IP process, while delaying endothelial cell necrosis, does not mitigate its extent; however, it effectively inhibits the intricate process of apoptosis in Human Umbilical Vein Endothelial Cells (HUVEC) subjected to hypoxia and reoxygenation. In addition, the IP process substantially reduces, by approximately 3-4 times, the loss of cell membrane integrity in HUVEC between 2 and 8 hours of reoxygenation. However, this protective effect is transient, as the percentage of viable ECs, both conditioned and unconditioned, becomes comparable after 24 hours of reoxygenation. Hence, the conditioning process temporarily retards but does not arrest the decline in cell membrane integrity in ECs. The IP process did not prevent the increase in bax expression during reoxygenation and did not alter bax expression after 8 hours compared to non-conditioned cells. However, after 24 hours, conditioning led to a significant reduction in bax expression compared to HUVEC not subjected to IP. Throughout the reoxygenation period, conditioned cells exhibited a significant increase in bcl-2 expression compared to control and hypoxia-exposed cells, resulting in a considerable enhancement of intracellular protective potential, as indicated by the bcl-2/bax ratio. Longer hypoxia was associated with lower values of bcl-2/bax ratio in EC subjected to HR or IP. In conclusion, IP delays, without reducing, the extent of HR-induced EC necrosis but significantly inhibits their multi-level evaluated apoptosis.
General concept comments
· The strength of this article is to reveal that the preconditioning process delays, without reducing, the extent of necrosis in endothelial cells exposed to hypoxia and reoxygenation. However, it significantly inhibits their multi-level evaluated apoptosis.
Questions
In lines74-77: the author mentioned that in an in vivo rat heart I/R model, it has also been demonstrated that a single 5-minute preconditioning stimulus preceding a 30-minute ischemic period significantly reduces the percentage of apoptotic cells in the threatened infarct zone, from 29% in the control group to 3.5% in the preconditioned tissue.
The in vitro HR model is intended, with some simplification, to imitate the I/R processes occurring in vivo, So I would like to ask a question:
Question 1 Why did this study design the ischemic preconditioning (IP) induced by 20 minutes of hypoxia followed by 20 minutes of reoxygenation? If this brief episode of ischemia is effective in inducing conditioning mechanisms, please show the evaluation of the effectiveness of this IP process.
Comment: Please have more discussion about the e-NOS and i-NOS expression. The i-NOS expression was not detected, Does it mean that the inflammation process is not occurring?
Author Response
Thank you for the positive feedback and all comments.
- Why did this study design the ischemic preconditioning (IP) induced by 20 minutes of hypoxia followed by 20 minutes of reoxygenation? If this brief episode of ischemia is effective in inducing conditioning mechanisms, please show the evaluation of the effectiveness of this IP process.
Different IP experimental protocols have been tested in cell culture so far. Zhou et al. showed that 10-minute of anoxia and 10-minute of reoxygenation protected coronary endothelial cells against anoxia and reoxygenation injury (Zhou et al Circ Res. 1996). Moreover, IP with 1-hour anoxia and 1-hour reoxygenation of cultured rat aortic endothelial cells prevented reoxygenation-induced, free radical-mediated expression of ICAM-1 involving activation of protein kinase C and production of nitric oxide and free radicals, and this was associated with a lesser adhesion of neutrophils to endothelial cells (Beaucham et al. Circulation 1999). In turn, in vivo IP with 10-min occlusion and reperfusion preceding 60-min ischemia and 120 min of reperfusion reduced endothelial dysfunction of coronary arterioles (DeFily et al. Am J Physiol. 1993). Also, the results of our unpublished preliminary experiments showed that IP induced by 20, 30 or 60 minutes of hypoxia with respective 20, 30 or 60 minutes of reoxygenation reduced LDH release following prolonged 6-hour hypoxia and 6-hour of reoxygenation. Therefore, in this study IP protocol with 20 minutes of hypoxia followed by 20 minutes of reoxygenation was applied.
Discussion section has been updated.
- Please have more discussion about the e-NOS and i-NOS expression. The i-NOS expression was not detected. Does it mean that the inflammation process is not occurring?
The key role in the development of reperfusion-induced apoptosis is attributed to the interaction between blood cells and endothelium [Nakamura et al. Cardiovasc Res 2000] mediated by the cytokines and reactive oxygen species released from activated neutrophils, endothelial cells and myocytes. Endothelial cells in the studied in vitro model were not subject to such interaction with blood cells, including neutrophils, what may be associated with the lack of induction of i-NOS expression. However, based only on the lack of i-NOS expression, it is difficult to answer whether the inflammatory process in cultured HUVEC was not activated the more that the expression of Cox-2 increased following 2-hour hypoxia and 24-hour reoxygenation and even more as a consequence of IP process. Basically, cox-2 expression accompanies harmful processes such as hypersensitivity, cancer, sepsis, apoptosis and also inflammatory process. On the other hand, the constant prostaglandins production by Cox-2 was found to be associated with late IP cardioprotection [Bolli et al. Cardiovasc Res 2022] and may also inhibit apoptosis process in various cells [von Knethen et al. FASEB J 1997].
Our in vitro model is a static cell culture in which endothelial cells are not exposed to the flowing medium but only incubated in fluid. Therefore, in these conditions endothelial cells are unable to respond to shear stress. In vitro, e-NOS expression is induced by shear stress, cyclic deformation, and oxidative stress while in-vivo after exercise and after agonist-induced endogenous NO release causing vasodilation [Govers et al. Am J Physiol 2001]. On the other hand, tissue hypoxia inhibits the expression of more than 95% of genes as compared to the baseline, and only a few genes increase their expression [Levy AP: Regulation of gene expression by hypoxia. Heart physiology and pathophysiology. Fourth edition, Academic Press 2001]. McQuillan et al. [Am J Physiol. 1994] showed that in endothelial cells underwent hypoxia, e-NOS expression was reduced by 40-60% and this effect lasted for at least 48 hours. In turn, 24-hour anoxia followed by 3-hour reoxygenation of human endothelial coronary cells was associated with significant reduction of the expression, protein concentration and activity of e-NOS [Li et al. Cardiovasc Res. 1999]. In our experiment, after 24 hours of reoxygenation, there was no change in e-NOS expression in the control group indicating a stable genotype of the endothelium despite the lack of shear stress, while the reduction of e-NOS expression in hypoxia/reoxygenation was prevented by the IP process.
Discussion section has been updated.